# Brick-by-Brick: Combinatorial Construction with Deep Reinforcement Learning

**Hyunsoo Chung**[*1]    **Jungtaek Kim**[*1]    **Boris Knyazev**[23]    **Jinhwi Lee**[14]
**Graham W. Taylor**[23]    **Jaesik Park**[1]    **Minsu Cho**[1]

[1]POSTECH    [2]University of Guelph    [3]Vector Institute    [4]POSCO
{hschung2,jtkim}@postech.ac.kr

## Abstract

Discovering a solution in a combinatorial space is prevalent in many real-world problems but it is also challenging due to diverse complex constraints and the vast number of possible combinations. To address such a problem, we introduce a novel formulation, *combinatorial construction*, which requires a building agent to assemble unit primitives (i.e., LEGO bricks) sequentially – every connection between two bricks must follow a fixed rule, while no bricks mutually overlap. To construct a target object, we provide incomplete knowledge about the desired target (i.e., 2D images) instead of exact and explicit volumetric information to the agent. This problem requires a comprehensive understanding of partial information and long-term planning to append a brick sequentially, which leads us to employ reinforcement learning. The approach has to consider a variable-sized action space where a large number of invalid actions, which would cause overlap between bricks, exist. To resolve these issues, our model, dubbed *Brick-by-Brick*, adopts an action validity prediction network that efficiently filters invalid actions for an actor-critic network. We demonstrate that the proposed method successfully learns to construct an unseen object conditioned on a single image or multiple views of a target object.

## 1    Introduction

A combinatorial space, typically characterized by discrete variables and their combinations, often induces interesting yet challenging problems such as traveling salesperson and minimum spanning tree [20, 6]. The main challenges lie in the vast number of possible combinations as well as complex constraints imposed on them. In a similar spirit, we suggest a novel problem formulation, *combinatorial construction*, that focuses on the real-world construction procedure. Given only incomplete target information (i.e., 2D images or multiple views of a target object) [25, 12], an agent sequentially assembles unit primitives (i.e., LEGO bricks). The proposed formulation is combinatorial since it engages repetitive placement of primitives, which leads to a large number of available solutions. Distinct property of our proposed formulation, however, is that the agent must build the solution incrementally by adding on to the partial solution. Specifically, a brick, which is a unit primitive of the object of interest, is placed on a discrete space by connecting to one of the previously assembled bricks. In addition, every connection between two bricks must follow a fixed rule while no bricks mutually overlap. Each assembly (i.e., action) executed by the agent is, thus, modeled as selecting one of the feasible connections to place a new brick.

The problem we introduce closely depicts how humans understand an object and adapt the acquired knowledge to a downstream task. Humans naturally analyze a 3D object by picturing its part-by-part

---

[*]Equal contribution.

35th Conference on Neural Information Processing Systems (NeurIPS 2021).

Table 1: Analysis of recent studies in terms of state representation, supervision, conditioning, target objects, and action validation. CE and IoU stand for cross-entropy and intersection over union with respect to volumetric comparisons. Direct forwarding (denoted as direct), sampling & checking (denoted as sampling), and our pretrained action validity prediction network (denoted as pretrained) indicate respective strategies that filter invalid actions; see the corresponding section for their details.

| Method | State | Supervision | Conditioning | Target | Action Validation |
|---|---|---|---|---|---|
| Hamrick et al. [11] | Image | Task-dependent | N/A | 2D | Direct |
| Bapst et al. [3] | Object/Image | Task-dependent | Object and/or image | 2D | Direct |
| Kim et al. [19] | Set | Overlap | Exact target volume | 3D | Sampling |
| Thompson et al. [38] | Graph | Step-wise CE | One-hot class info. | 3D | Direct |
| Brick-by-Brick ($B^3$, ours) | Graph/Image | IoU | Image or set of images | 3D | Pretrained |

decomposition and consequently grasp a rich semantic understanding [15, 22]. In various fields, they utilize an inherent ability to decompose objects to effectively solve challenging tasks such as object classification [17], robot grasp planning [2], and part segmentation [31, 28]. Likewise, humans exploit this ability to solve the inverse problem – combinatorial construction. Given a desired object to be constructed and no strong supervision (i.e., ordered step-by-step instructions), humans can often still manage to build a valid target object by carefully planning or, sometimes improvising, the sequence of actions. Our environment, which corresponds to the proposed problem, is designed to learn and test such behavior with only partial information of the desired target available to the agent.

Successfully constructing an object in our setup requires a comprehensive understanding of incomplete target information with the current structured state of assembled bricks and long-term planning to append each brick efficiently. These requirements, along with the absence of sequence-level supervision, incentivize us to devise a reinforcement learning (RL) approach [3, 37]. In this domain, however, we must carefully handle both an indefinite action space and the existence of many invalid actions when applying RL [43]. In particular, both defining an action space that varies by the number of assembled bricks and distinguishing an invalid action that would cause overlap with other existing bricks quickly become intractable as more bricks are placed. To resolve the aforementioned issues, our model, dubbed Brick-by-Brick ($B^3$), adopts an action validity prediction network that filters invalid actions to an actor-critic network. In addition to the novel RL formulation, we use graph-structured representation of the brick combination to interpret the assembling process as a sequential graph generation process.

Overall, we summarize our contributions as follows:

(i) We propose a novel problem formulation, combinatorial construction, that closely resembles a real-world object construction process that engages repetitive placement of components;

(ii) We design an RL agent for combinatorial construction, dubbed Brick-by-Brick ($B^3$), to effectively address both a growing action space and a vast set of invalid actions;

(iii) We implement the corresponding environment based on OpenAI Gym and introduce new novel evaluation scenarios that vary by their incomplete partial target information.

## 2   Combinatorial Construction

To formulate the combinatorial construction problem, we start by defining a unit primitive that is used to construct a 3D object and an action space that determines where to assemble the next primitive.

As a unit primitive, we utilize a $2 \times 4$ brick, which has eight studs and their fit cavities. This design choice yields a consistently varying action space, implying that if we add one brick to the current state of brick combination, we can efficiently define the next action space. We want to emphasize that with only six $2 \times 4$ bricks, we can create *915,103,765 combinations* [9]. Accordingly, our choice of the primitive does not make our problem a trivial task; instead, every decision of where we place the next primitive can deteriorate the quality of the final result because there exists a plethora of wrong paths.

With our specific choice of unit primitives, we can define an action space for determining the next action and evaluating the future states. However, since every assembly step gradually expands the

action space, naïve approaches to defining the growing action space are not appropriate for our problem; an action space with redundant actions [43] is not applicable due to a varying action space, and an action sampling approach [13] is also not suitable due to nominal or invalid actions. Thus, we define a *successive* action space composed of a two-step decision: (i) choosing a *pivot brick* and (ii) choosing an *offset* from the pivot brick.

Before explaining a pivot brick, we first assume a simplified assembly scenario that follows an Eulerian path[2] – a new brick is always placed by connecting to the last assembled brick. This enables us to define a finite action space though most actions are infeasible to perform with the Eulerian path. To broaden the search space by generalizing the Eulerian path, one of previously assembled bricks is chosen as a pivot brick. Then, $B^3$ decides an offset from the pivot brick, which describes how the next brick is placed relative to the pivot. Thanks to the homogeneous brick type, the number of available offsets is finite and consistent where we do not consider the validity of such offsets in a certain state – for $2 \times 4$ bricks, there exist a maximum of 92 available offsets.

Due to the disallowance of overlap between bricks, our agent must consider *invalid actions* among available actions during assembly. Identifying the validity of actions from the current brick combination becomes intractable as more bricks are placed; the complexity of this process is $\mathcal{O}(|A_{\text{off}}|t^2)$, where $A_{\text{off}}$ is an action space for offsets and $t$ is the cardinality of assembled bricks at a given step; see the supplementary material for qualitative results on this complexity. Such expensive overhead for validation naturally leads us to adopt an action validity prediction network that learns to identify invalid actions with a single forward pass.

As described in Section 1 and this section, our problem has interesting but challenging characteristics derived from the assumptions on discrete placement, a connectivity rule, disallowance of overlap, and ultimately invalid actions. We, therefore, present a comparison to other existing studies in terms of state, supervision (or a reward function), conditioning, target objects, and action validation, as shown in Table 1. Compared to the previous work [11, 3, 19, 38], our method $B^3$ constructs an object in 3D with incomplete target information and pretrained validity prediction component; see Sections 4 and 5 for a detailed description.

# 3   Brick-by-Brick

In this section, we briefly introduce the definition of RL and the corresponding framework for combinatorial construction, where an agent places a brick sequentially. We then explain the details of our model that learns to select appropriate actions given only partial information of the desired target so that the assembled 3D object resembles the target. Moreover, to cope with a vast number of invalid actions in the process of assembly, we propose an action validity prediction network. See Figure 1 for the overall pipeline of our method $B^3$.

**Definition.**   In a standard RL framework, there exists an agent that interacts with an environment by iteratively making decisions given an observation of the environment. This follows general decision making procedure of a Markov decision process (MDP), where a transition function satisfies the Markov property, i.e., $p(s_{t+1}|s_0, s_1, \ldots, s_t, a_t) = p(s_{t+1}|s_t, a_t)$, where $s_t$ and $a_t$ are a state and an action at timestep $t$, respectively.

In our problem setting, we only consider a finite horizon MDP formally defined as a tuple of $(S, A, P, R, \gamma)$, where $S = \{s_t\}$ is a set of states, $A = \{a_t\}$ a set of actions, $R : S \times A \to \mathbb{R}$ a reward function, $P : S \times A \to S$ a transition function, and $\gamma \in [0, 1)$ a discount factor. The goal of the agent is to learn a policy $\pi(a_t|s_t)$ that maximizes the expected future cumulative reward. We introduce our detailed MDP formulation for combinatorial construction in the following sections.

## 3.1   Problem Formulation

Given target information $\mathcal{T}$, the agent aims to construct a 3D target object $\mathbf{T}$ by assembling bricks sequentially, one brick for each $t$-th step. Each $t$-th brick is represented by its pose $(\mathbf{x}_t, d_t)$, where $\mathbf{x}_t \in \mathbb{Z}^3$ is the center coordinate of the brick in a 3D space and $d_t \in \{0, 1\}$ denotes one of two possible directions, meaning that its longer axis is aligned along either $x$ axis or $y$ axis in the 3D space.

---

[2]It is a path that visits all the vertices without revisiting the edges visited before.

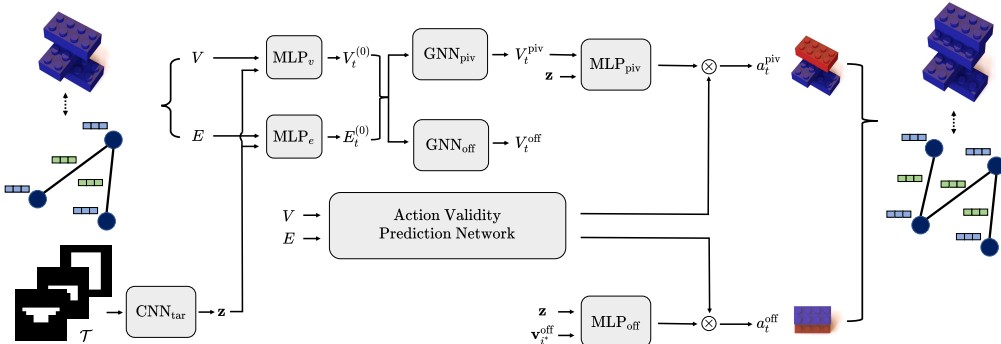

Figure 1: Overview of our proposed method $\text{B}^3$. A state input $s_t = (G_t, \mathcal{T})$ is embedded and passed through CNNs, GNNs, and MLPs to predict an action $a_t$ that consists of a pivot brick indicator $a_t^{\text{piv}}$ and an offset from the pivot brick $a_t^{\text{off}}$. Moreover, an action validity prediction network helps to filter invalid actions from the action space where we determine the next brick to assemble. The red brick in both actions of $a_t^{\text{piv}}$ and $a_t^{\text{off}}$ indicates the chosen brick. See Section 3 for the details.

**Target Information.** We are given as target information $\mathcal{T}$ a set of binary images of a target object, which may correspond to incomplete and partial information about the target. In practice, this setup with partial information is more realistic than accessing full information of a 3D target shape. Our task thus is to create a sequence of unit bricks by inferring a target object from abstract information in a combinatorial manner.

**State.** Each $t$-th state $s_t$ of the MDP is represented by a tuple of a directed graph $G_t$ composed of $t$ bricks and target information $\mathcal{T}$, i.e., $s_t = (G_t, \mathcal{T})$. The graph is defined as $G_t = (V_t, E_t)$, where $V_t = \{\mathbf{v}_i\}_{i=1}^t$ is a set of $t$ bricks, i.e., $\mathbf{v}_i = (\mathbf{x}_i, d_i) \in \mathbb{Z}^4$, and $E_t = \{\mathbf{e}_{ij}\}_{i,j=1}^t$ is a set of the offset vectors between two connected nodes, i.e., $\mathbf{e}_{ij} = (\mathbf{x}_i - \mathbf{x}_j, d_i \oplus d_j) \in \mathbb{Z}^4$. Note that nodes are connected by edges according to sequential actions and relative offsets in pose are used for edge features in order to induce translational and orientational equivariance. Since all the edges are bi-directional, we omit the arrows when displaying graphs in Figure 1.

**Action.** We define a successive action space of choosing the pivot brick first and the corresponding offset next. Formally, with $t$ bricks assembled, we define an action $a_t = (a_t^{\text{piv}}, a_t^{\text{off}})$ where $a_t^{\text{piv}}$ is to select a pivot brick and $a_t^{\text{off}}$ is to select an offset with respect to the pivot brick. The pose of the next brick is then $(\mathbf{x}^{\text{piv}} + \Delta\mathbf{x}, (d^{\text{piv}} + \Delta d) \mod 2)$, where $(\mathbf{x}^{\text{piv}}, d^{\text{piv}})$ and $(\Delta\mathbf{x}, \Delta d)$ are determined by $a_t^{\text{piv}}$ and $a_t^{\text{off}}$, respectively. In choosing actions $a_t^{\text{piv}}$ and $a_t^{\text{off}}$, we exclude invalid actions: (i) choosing a pivot brick near which no additional brick can be placed and (ii) choosing an offset for the next brick that overlaps with existing bricks. To select a valid action in the action space with a vast number of such invalid ones, we predict invalid actions in advance using the action validity prediction network and exclude them from action candidates preemptively; we mask out all the probabilities of invalid actions, re-normalize the distribution over actions $a_t$, and sample an action from the distribution.

**Transition Function.** Given state $s_t$ and action $a_t$, our transition function $p(s_{t+1}|s_t, a_t)$ is designed to determine the next state $s_{t+1}$ by deterministically updating $s_t$ based on $a_t$. The node of the new brick $\mathbf{v}_{t+1}$ is created so that $V_{t+1} = V_t \cup \{\mathbf{v}_{t+1}\}$. The edges between the new brick and existing bricks in physical contact via studs are created so that $E_{t+1} = E_t \cup \{\mathbf{e}_{(i)(t+1)}\}_{i \in \mathcal{N}_{t+1}} \cup \{\mathbf{e}_{(t+1)(i)}\}_{i \in \mathcal{N}_{t+1}}$, where $\mathcal{N}_{t+1}$ denotes the set of bricks in direct contact with the new brick $\mathbf{v}_{t+1}$. As the result, the graph in the state $s_{t+1}$ is updated to $G_{t+1} = (V_{t+1}, E_{t+1})$.

**Reward Function.** In contrast to the tasks where a direct reward evaluation is readily available, it is not trivial to quantify the object assembled by combinatorial construction, especially, in the context of graph generative model [26]. To mitigate such an issue, we exploit the property of a voxel

representation. Given a desired object, we first create voxels in a closed space and determine the occupancy of voxels with a target object, after normalizing it to the bottom center of voxels. We then transform the combination of currently assembled bricks into the occupancy of the voxels and measure the overlap with the target object:

$$\Delta\text{IoU}(\mathbf{C}_t, \mathbf{T}) = \frac{\text{vol}(\mathbf{C}_t \cap \mathbf{T})}{\text{vol}(\mathbf{C}_t \cup \mathbf{T})} - \frac{\text{vol}(\mathbf{C}_{t-1} \cap \mathbf{T})}{\text{vol}(\mathbf{C}_{t-1} \cup \mathbf{T})}, \tag{1}$$

where $\mathbf{C}_t$, $\mathbf{C}_{t-1}$, and $\mathbf{T}$ are the occupied voxels at timestep $t$, timestep $t-1$, and a desired target, respectively. In addition, $\text{vol}(\cdot)$ is a function that measures a volume. The step-wise reward function is then $\Delta\text{IoU}$ if the new brick overlaps at least 50% with the occupied voxels of target object and 0 otherwise. Consequently, our agent will learn sequential placement of bricks to construct the object, without explicit supervision by maximizing Equation (1), as will be described in the subsequent section.

## 3.2 Sequential Construction

In this section, we describe how we process $G_t$ and $\mathcal{T}$, which comprise a state, with different types of neural networks such as convolutional neural networks and graph neural networks. The overview of this construction procedure is illustrated in Figure 1.

**Node and Target Embeddings.** Given a state $s_t = (G_t, \mathcal{T})$ where $G_t = (V_t, E_t)$, we first use a convolutional neural network (CNN) to extract features $\mathbf{z}$ from the target:

$$\mathbf{z} = \text{CNN}_{\text{tar}}(\mathcal{T}). \tag{2}$$

If the partial information is given as a set of images, the feature $\mathbf{z}$ is obtained by first applying CNN to each image separately and then concatenating the outputs to a single vector.

For node and edge features, an MLP embeds them with the target feature $\mathbf{z}$:

$$\mathbf{v}_i^{(0)} = \text{MLP}_v([\mathbf{v}_i, \mathbf{z}]), \quad \text{and} \quad \mathbf{e}_{ij}^{(0)} = \text{MLP}_e([\mathbf{e}_{ij}, \mathbf{z}]), \tag{3}$$

for all $i, j \in \{1, \ldots, t\}$, where $[,]$ and $^{(0)}$ denote concatenation and the first layer, respectively.

Equations (2) and (3) can be viewed as pre-processing inputs to feed in a graph neural network (GNN). Inspired by [4], we apply a variant of graph networks (GNs) in which a global graph feature is omitted. At the $\ell$-th layer of GNNs, we update edge features, aggregate the messages for each node, and update node features:

$$\mathbf{e}_{ij}^{(\ell+1)} = \text{MLP}_e^{(\ell)}\left([\mathbf{v}_i^{(\ell)}, \mathbf{v}_j^{(\ell)}, \mathbf{e}_{ij}^{(\ell)}]\right), \tag{4}$$

$$\mathbf{m}_i^{(\ell)} = \sum_{j \in \mathcal{N}_i} \text{aggregate}\left(\mathbf{e}_{ij}^{(\ell+1)}\right), \tag{5}$$

$$\mathbf{v}_i^{(\ell+1)} = \text{MLP}_v^{(\ell)}\left([\mathbf{v}_i^{(\ell)}, \mathbf{m}_i^{(\ell)}]\right), \tag{6}$$

where $\mathcal{N}_i$ is the neighborhood nodes of $\mathbf{v}_i$, and aggregate$(\cdot)$ is the aggregation function that computes a message for each node by aggregating the features of its neighboring nodes. Note that $\text{MLP}_v$, $\text{MLP}_e$, $\text{MLP}_v^{(\ell)}$, and $\text{MLP}_e^{(\ell)}$ have their own learnable parameters.

**Action Selection.** In order to enrich representations for predicting $a_t$, we employ two separate GNNs: $\text{GNN}_{\text{piv}}$ and $\text{GNN}_{\text{off}}$, with $L$ layers in total, to produce sets of node embeddings, $V_t^{\text{piv}}$ and $V_t^{\text{off}}$ for pivots and offsets:

$$V_t^{\text{piv}} = \text{GNN}_{\text{piv}}(V_t^{(0)}, E_t^{(0)}), \quad \text{and} \quad V_t^{\text{off}} = \text{GNN}_{\text{off}}(V_t^{(0)}, E_t^{(0)}), \tag{7}$$

where $V_t^{(0)} = \{\mathbf{v}_i^{(0)}\}_{i=1}^t$ and $E_t^{(0)} = \{\mathbf{e}_{ij}^{(0)}\}_{i,j=1}^t$ are the sets of node and edge features obtained by Equation (3). Each layer of both GNNs updates node and edge features using Equations (4), (5), and (6). Finally, the set of node and edge features, $V_t^{\text{piv}}$ and $V_t^{\text{off}}$, along with a target feature $\mathbf{z}$ are used to decide the next action $a_t^{\text{piv}}$ and $a_t^{\text{off}}$:

$$p\left(a_t^{\text{piv}}\right) = \sigma\left(\text{MLP}_{\text{piv}}([V_t^{\text{piv}}, \mathbf{z}])\right), \quad \text{and} \quad p\left(a_t^{\text{off}}\right) = \sigma\left(\text{MLP}_{\text{off}}([\mathbf{v}_{i^*}^{\text{off}}, \mathbf{z}])\right), \tag{8}$$

where $\sigma$ is a softmax function and $\mathbf{v}_{i^*}^{\text{off}}$ is the node feature selected by the index $i^*$ of $a_t^{\text{piv}}$.

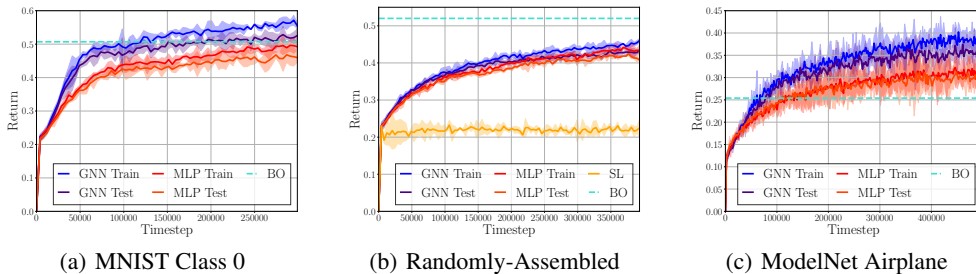

(a) MNIST Class 0       (b) Randomly-Assembled       (c) ModelNet Airplane

Figure 2: Episode return vs. timestep in different setups. The return values in training and test episodes are reported by repeating 3 times with different seeds.

**Action Validation.** To tackle the issue of a vast number of invalid actions, we propose to learn an action validity prediction network. Previous work adopts direct forwarding [3, 38] or sampling & checking [19]. In the direct forwarding approach [3, 38], the agent directly selects an action without any prior processing. This typically suffers from the early termination of episodes since the sequence of actions terminates with a deadlock as soon as the agent selects an invalid action. In the sampling & checking approach [19], valid actions are collected by sampling a random set of actions and checking the validity for each of them directly. This requires a high cost of iterative checking and the actions are limited to a small collection of checked actions.

In contrast to these approaches, we train a separate module to predict a large set of valid actions, enabling the agent to sufficiently explore the action space. We train a GNN, of which the node-wise output predicts validity confidences for its candidate actions, i.e., candidate actions for the corresponding brick. The structures of both pivot and offset validity prediction networks are identical to the networks described in Equation (8), but the last activation is a sigmoid function and no target feature $\mathbf{z}$ is used. Importantly, these networks can be pretrained by the ground-truth action validity, which is obtained from randomly-assembled objects, and such a pretrained network can be used in training an actor-critic network, *without re-training*.

Note that, if our action validity prediction network fails to filter invalid actions and one of such actions is selected by the agent, the corresponding episode is terminated. Unlike the direct forwarding approach, training the agent with the action validity prediction network does not suffer from early termination as the validity prediction network masks out the majority of invalid actions.

**Training.** We adopt the proximal policy optimization (PPO) algorithm [35], which is one of the state-of-the-art on-policy algorithms. In particular, we optimize the clipped surrogate objective over parameters $\boldsymbol{\theta}$:

$$\mathcal{L}(\boldsymbol{\theta}) = \mathbb{E}\left[\min(r_t(\boldsymbol{\theta})\hat{A}_t, \text{clip}(r_t(\boldsymbol{\theta}), 1 - \epsilon, 1 + \epsilon)\hat{A}_t)\right], \tag{9}$$

where $r_t(\boldsymbol{\theta})$ is a probability ratio between the previous and updated policy, clip is a clipping function between the second and the third arguments, and $\hat{A}_t$ is an advantage function [34]. To calculate the advantage of a state $s_t$, our model employs a value network, $\text{MLP}_{\text{val}}([\mu(V_t^{\text{piv}}), \mu(V_t^{\text{off}}), \mathbf{z}])$, where $\mu(\cdot)$ is a global average function over instances in a given set.

## 4 Experimental Results

We evaluate our image-conditioned 3D object assembly of B$^3$ in three scenarios: (i) MNIST construction, (ii) randomly-assembled object construction, and (iii) ModelNet construction.

For an evaluation metric, we measure the episode return or IoU between the constructed object and the desired target at the end of each episode:

$$\text{IoU}(\mathbf{C}_N, \mathbf{T}) = \frac{\text{vol}(\mathbf{C}_N \cap \mathbf{T})}{\text{vol}(\mathbf{C}_N \cup \mathbf{T})}, \tag{10}$$

where $N$ is the total number of bricks and $\mathbf{T}$ is the voxel representation of the target object. The maximum number of bricks to be placed depends on $\mathcal{T}$ and is pre-defined. After exhausting the brick

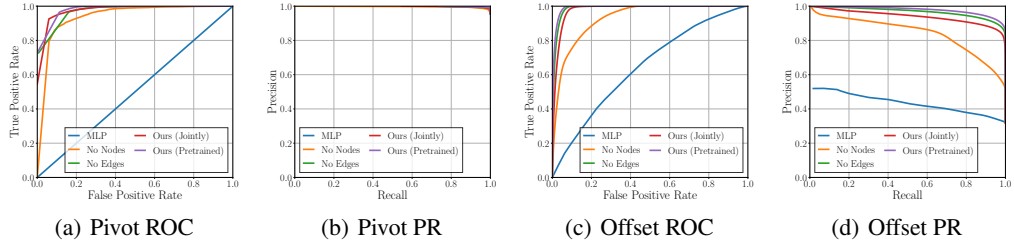

| (a) Pivot ROC | (b) Pivot PR | (c) Offset ROC | (d) Offset PR |

Figure 3: ROC and PR curves for the action validity prediction network. All the results are measured using the test dataset of randomly-assembled objects.

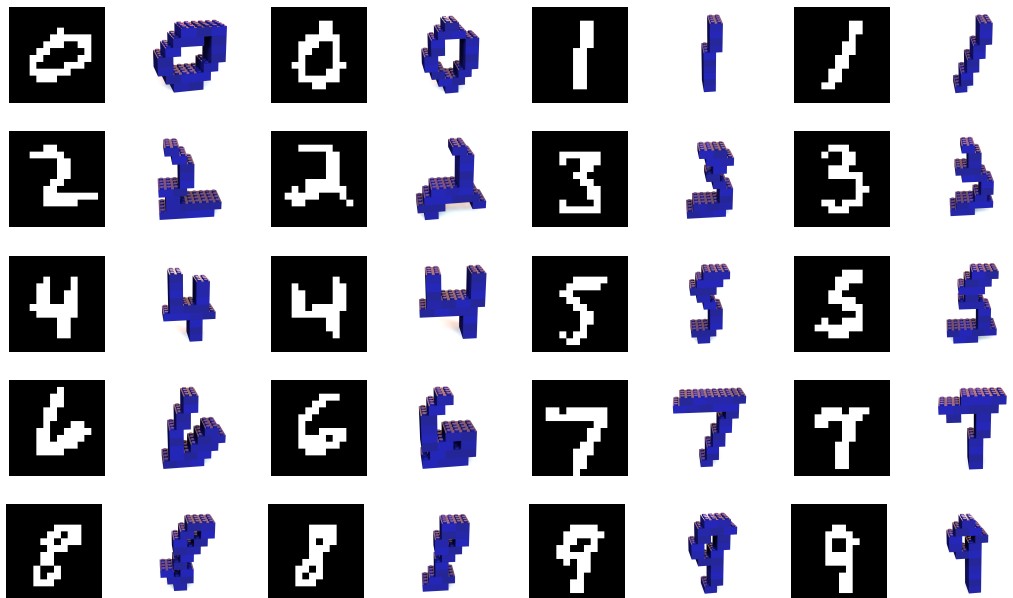

Figure 4: Qualitative results on MNIST construction. Our model is separately trained on each class, and target images are unseen during training.

budget, we terminate the episode and compute the final IoU. Unless otherwise specified, we report the average performance over 3 random seeds, each of which is trained for a fixed timestep budget.

To show the effectiveness of our method, we first analyze our action validity prediction network and test other baseline methods and $B^3$ in different scenarios. In all construction tasks, we compare $B^3$ to the MLP-based model where all GNNs are replaced with MLPs, and to the Bayesian optimization-based approach (BO) that sequentially optimizes the step-wise reward in terms of IoU to search for an optimal construction sequence. As presented in Table 1, BO uses exact volumetric information for both training and test target objects because it cannot assemble an object with only partial information. For each scenario, the episode returns of the BO model are averaged over both training and test datasets. In addition, we compare $B^3$ to the supervised learning method trained with the cross-entropy loss between predicted and ground-truth sequences, specifically, in the randomly-assembled object construction. Since sequence-level supervision is used, the performance of the supervised learning method is only measured on the test dataset. Details can be found in the supplementary material.

**Action Validity Prediction Network.** We test our action validity prediction network by creating training and test datasets. The training dataset is composed of 200,000 brick combinations and their ground-truth action validity, and the test dataset is composed of 30,000 brick combinations and their ground-truth action validity. Importantly, the range of the size of a brick combination in the training dataset is $[1, 20]$, and the range in the test dataset is $[1, 30]$. While the test dataset contains larger brick combinations than the training dataset, the performance of the action validity prediction network in

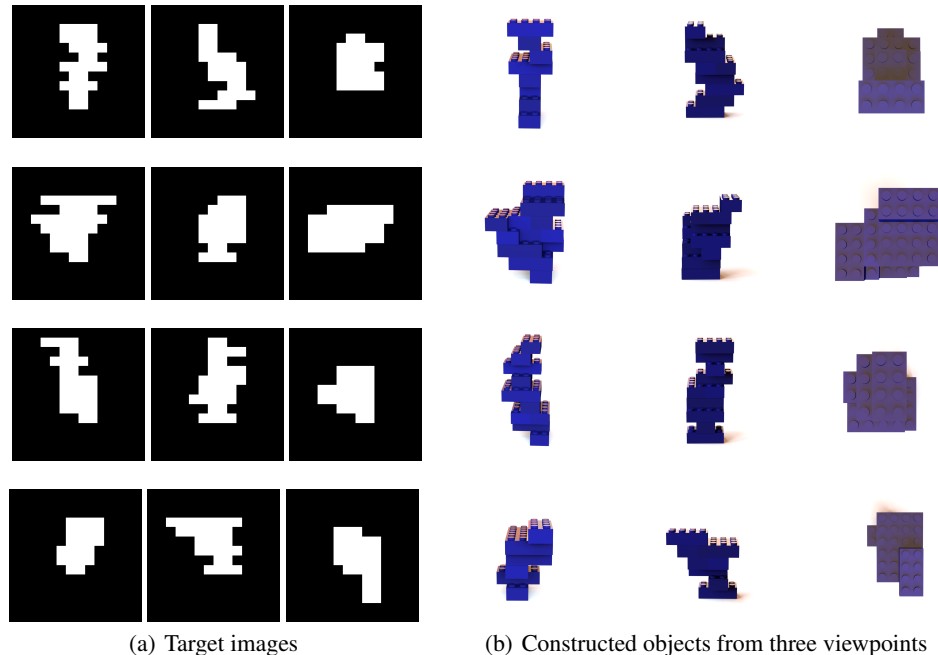

(a) Target images          (b) Constructed objects from three viewpoints

Figure 5: Qualitative results on randomly-assembled object construction. Targets are obtained from the test dataset, and each row represents a pair of target information and constructed object.

terms of precision and recall is satisfactory, predicting reliable validity confidences even for actions in unseen ranges, as presented in Figure 3. Our GNN outperforms MLP as well as GNN baselines, which do not have either node features or edge features. In addition, the pretrained network, which is reusable in different scenarios, is slightly better than the action validity prediction network that is jointly trained with training episodes. See the supplementary material for more details of the action validity prediction network.

**MNIST Construction.** In each episode, an agent is provided with an image from the MNIST dataset and is provided to create a 3D object resembling the digit target. Similar to [32], we binarize the MNIST dataset to convert a real-valued number to either 0 or 1, for brevity of the calculation of IoU. To create a 3D target object with a 2D MNIST image, we first rescale an image to half of the original size and then expand an image along the channel dimension, in order to assemble with $2 \times 4$ bricks, i.e., an image of size $28 \times 28$ is transformed to a 3D object of size $14 \times 14 \times 4$. Furthermore, we limit possible offset candidates to 6 different types of which the values according to the channel dimension are fixed to the same value. Training and test datasets are established by choosing one of the ten classes in the binarized MNIST and splitting images from that class. In particular, 500 images from one of available classes are chosen, further divided into 400 samples for a training dataset and 100 samples for a test dataset.

Due to a space limit, we only report the average reward performance on class 0 in Figure 2(a). Results for other classes are available in the supplementary material. The gap between the training and test sets on episode returns is marginal, which implies that our model $B^3$ generalizes to unseen targets well. In addition, both our training and test results are better compared to both the BO and MLP-based models. We visualize constructed objects for the test dataset of classes 0 through 9 in Figure 4. More qualitative results are also provided in the supplementary material. In general, our agent successfully constructs objects of unseen instances. This can be understood as our agent catches distinctive details in the target information and reflects in a construction process.

**Randomly-Assembled Object Construction.** Contrary to the experiments of MNIST construction, this task focuses on building objects that require more than one image to fully understand the structure. Accordingly, the agent must construct an object with three $14 \times 14$ images from different viewpoints, which are initially given as the target information. Objects in this experiment are artificially generated

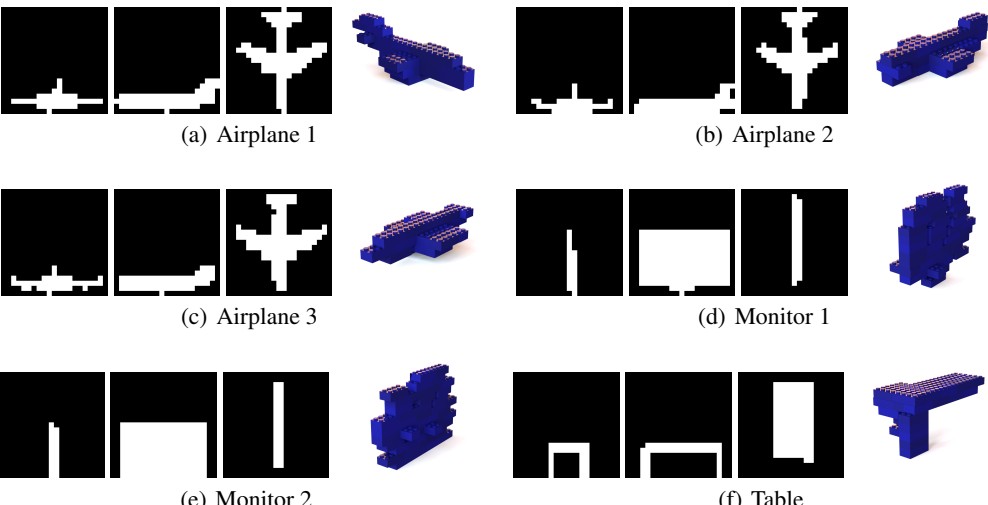

Figure 6: Results on ModelNet construction. Targets are obtained from the test dataset. The first three panels and the last panel of each figure show the target images and the constructed example.

by connecting bricks at random. The total number of bricks is also chosen randomly between 10 to 15. For available offset types, we only utilize connection types that occupy four or more studs and only allow a new brick to be placed on top of the pivot brick so that the resulting target becomes more distinguishable; see the supplementary material for the details. In this scheme, the total number of offsets is 16. Finally, we sample images from 800 target objects for a training dataset while 200 target objects are used for a test dataset.

As shown in Figure 2(b), our model achieves a return comparable to the MLP-based model while a slightly lower return compared to BO. We conjecture that this is due to the relatively small number of used bricks compared to other test suites. Nevertheless, our agent is still capable of associating the target object in 3D space from multiple images as illustrated in Figure 5. Our model learns to assemble bricks in a way that the resulting object successfully matches the initially given images, whereas the model trained with the supervised learning method does not generalize to unseen images of the test dataset. This clearly demonstrates the effectiveness of applying RL compared to learning with sequence-level supervision.

**ModelNet Construction.** Similar to randomly-assembled object construction, the agent is given 3 images of a realistic target from the ModelNet dataset [40]. To adjust the difficulty of this task, we find an object that is able to limit the maximum budget of bricks to less than 60. As a result, we choose airplane, monitor, and table categories from the ModelNet dataset. Moreover, we use offset types that connect with four or more studs and allow a new brick to be placed above and below the pivot brick. This task is the most challenging due to the excessive search space compared to MNIST construction and randomly-assembled object construction, and assesses the agent's ability to generate a real-world target object.

We provide training and testing curves for the airplane class in Figure 2(c); see the supplementary material for more results for monitor and table categories. Despite the difficulty raised from the large search space and long sequence, the result demonstrates that $B^3$ is capable of learning the construction process of real-world objects. BO with a limited budget achieves lower return compared to $B^3$ since the search space is too big to explore with limited computation. By comparing the constructed object to images of the desired target in Figure 6, it can be observed that $B^3$ generally captures overall shape of the target. Though, our model tends to struggle to catch fine-grained details such as wings of the airplane or legs of the table. However, for example, a table with only three legs (i.e., one leg missing) or two legs in a diagonal direction would perfectly match with the same three input views. It implies that if we provide more complete target information than three different views of target object, our agent can construct the target object more precisely; see the supplementary material for more detailed discussion on this limitation.

# 5 Related Work

In this section, we briefly cover related work on the task solved in this work.

**3D Object Generation.**  Following the studies on 2D object generation, e.g., the work by Dosovitskiy et al. [8], 3D object generation is often achieved in holistic manner [39, 1, 14, 41]. They generate a 3D object in a single feed-forward operation which limits exploitation of intermediate structures. Compared to these holistic methods, Kim et al. [19] propose an approach to tackle a combinatorial assembly problem by using Bayesian optimization [5], not a learning-based method. Unlike [19], Thompson et al. [38] apply a graph-structured generative model in the combinatorial 3D object generation task, by training to match a ground-truth sequence of LEGO bricks.

**Graph-based Reinforcement Learning.**  A common learning-based technique for creating a graph is to use one of various models such as recurrent neural networks [42, 24], adversarial networks [7], variational autoencoders [18, 33, 36], and Transformers [30]. Unlike these directions, Simm et al. [37] solve this idea of generating molecules with RL such that generated molecules are placed in the Cartesian coordinate. The key difference to our work is that we sequentially generate 3D shapes which have a much larger search space. Furthermore, Bapst et al. [3] show that an RL agent can learn physical construction in 2D space, and utilize rich visual information as well as a graph-structured representation, in order to define a state and a search space.

**Image-Conditioned Reinforcement Learning.**  Ganin et al. [10] propose an approach to synthesizing a program for 2D images when either an unconditional or conditional scenario is assumed. Their method generates an image by sequentially conducting an action in the MuJoCo environment. Nair et al. [29] propose a goal-conditioned RL approach the goal of which is provided by visual information. Huang et al. [16] suggest a method to paint a palette with stokes where a target image is conditionally given, by utilizing an RL algorithm.

**Brick Assembly Optimization.**  The brick assembly problem satisfying pre-defined constraints is a longstanding topic in computer graphics. Lee et al. [23] tackle LEGO brick layout optimization by a genetic algorithm. Similarly, Luo et al. [27] solve building sculptures safely with LEGO brick by stability aware refinement. Zhang et al. [44] propose the method for generating component-based building instructions that is safe based on segmentation models. Kozaki et al. [21] tackle a similar problem of brick assembly from images with the octree voxel-based model. This line of research tends to focus on directly utilizing the voxel representation of target object, instead of extracting the representation of target object from incomplete partial information.

# 6 Conclusion

We have proposed a novel problem formulation, combinatorial construction, which asks an agent to construct an object sequentially. We adopt RL by defining a state as graph-structured representation to express assembled bricks and their connections, where incomplete target information is given. In addition, we develop our algorithm with a successive action space that does not depend on the number of bricks already constructed and a reward function that measures overlap between a target object and the current state. Through extensive experiments, we demonstrate that our method can construct objects in various construction scenarios, and provide the analysis of our action validity prediction network.

## Acknowledgments and Disclosure of Funding

This work was supported by the IITP grants (No.2019-0-01906: AI Graduate School Program - POSTECH, No.2021-0-02068: AI Innovation Hub) funded by Ministry of Science and ICT, Korea and Samsung Electronics Co., Ltd (IO201208-07822-01). JK carried out this research during a research internship at the Vector Institute, and JK and HC equally contributed to this work. BK was funded by NSERC and the Ontario Graduate Scholarship. GWT and BK also acknowledge support from CIFAR and the Canada Foundation for Innovation. Resources used in preparing this research were provided, in part, by the Province of Ontario, the Government of Canada through CIFAR, and companies sponsoring the Vector Institute: `http://www.vectorinstitute.ai/#partners`. We also thank Hyeonwoo Noh for helpful discussions.

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
