# Supplementary Material for "Brick-by-Brick: Combinatorial Construction with Deep Reinforcement Learning"

**Hyunsoo Chung**[*1]    **Jungtaek Kim**[*1]    **Boris Knyazev**[23]    **Jinhwi Lee**[14]
**Graham W. Taylor**[23]    **Jaesik Park**[1]    **Minsu Cho**[1]

[1]POSTECH    [2]University of Guelph    [3]Vector Institute    [4]POSCO
{hschung2,jtkim}@postech.ac.kr

In this material, we first describe the importance of action validity prediction networks. Then, we introduce the details of the benchmarks, provide the model architecture, and present the additional experimental results, which are missing in the main article. Finally, we discuss limitations and societal impacts of our work in the last section.

## S.1  Action Validity Prediction Network

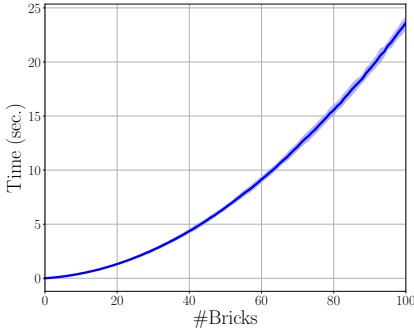

Figure s.1: Results of wall-clock time for computing the ground-truth action validity. We repeat 10 times and plot $\pm 1.96$ standard deviation.

Table s.1: Comparisons of action validation approaches.

| Method | Separate module | No access to action validity in test phase | Reusability |
|---|---|---|---|
| Direct forwarding | | ✓ | |
| Sampling & checking | ✓ | | |
| Ours (Jointly) | ✓ | ✓ | |
| Ours (Pretrained) | ✓ | ✓ | ✓ |

Compared to the construction cases with ground-truth action validity, the cases with our action validity prediction network are beneficial in terms of computational costs. We present the results of wall-clock time for computing the ground-truth action validity in Figure s.1. It shows that computing the action validity for a combination of 100 bricks needs more than 20 seconds. Moreover, we summarize the comparisons between possible action validation approaches as shown in Table s.1.

---

[*]Equal contribution.

35th Conference on Neural Information Processing Systems (NeurIPS 2021).

Table s.2: Results on predicting invalid actions by an action validity prediction network. Thresholds for deciding either valid or invalid actions are set to 0.5.

|  | **Pivot** | | | | **Offset** | | | |
|---|---|---|---|---|---|---|---|---|
|  | Training | | Test | | Training | | Test | |
|  | Precision | Recall | Precision | Recall | Precision | Recall | Precision | Recall |
| MLP | 0.9618 | **1.0000** | 0.9557 | **1.0000** | 0.5614 | 0.1410 | 0.5130 | 0.1398 |
| No Node | 0.9874 | 0.9895 | 0.9804 | 0.9869 | 0.8261 | 0.7518 | 0.7931 | 0.7344 |
| No Edge | 0.9947 | 0.9986 | 0.9850 | 0.9948 | 0.9199 | **0.9736** | 0.8897 | **0.9672** |
| Ours (Jointly) | 0.9881 | 0.9988 | 0.9809 | 0.9982 | 0.9001 | 0.9505 | 0.8674 | 0.9467 |
| Ours (Pretrained) | **0.9976** | 0.9987 | **0.9909** | 0.9944 | **0.9408** | 0.9709 | **0.9125** | 0.9661 |

As described in the main article, our action validity prediction network can be pretrained using the episodes obtained from the randomly-assembled object construction task and only requires a single forward pass to compute the action validity in inference time. In addition to these, we show the results on predicting invalid actions by an action validity prediction network in Table s.2. The results shown in Figure 3 and this table demonstrate that our pretrained network is effective in predicting action validity. For the jointly-trained validity prediction network, we assume that the oracle agent decides the next actions by obtaining them from the training dataset, which implies that all the 10,000 episodes in the training dataset are used to train the action validity prediction network with a single epoch.

## S.2   Details of Benchmarks

At the beginning of each episode, the agent starts with a single brick placed at the origin with the direction 0 regardless of the type of experiment it is being tested on. Specifically, the agent is given a graph with a single node feature of $[0, 0, 0, 0]$ and the edge feature matrix of zero values along with target information. All the hyperparameters are described in Tables s.3 and s.4. Below, we present the additional details distinctive for each benchmark.

### S.2.1   MNIST Construction

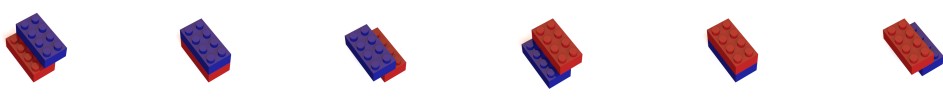

Figure s.2: Visualization of available offsets for MNIST construction.

Available offsets are visualized in Figure s.2. Since new brick (colored in dark blue) can be placed below the pivot brick (colored in red), the total number of offsets is 6.

Brick budget for each instance is set to 110% of the total number of the pixels that have value 1 in a target MNIST image.

### S.2.2   Randomly-Assembled Object Construction

Available offset types are illustrated in Figure s.3. Unlike the experiments of MNIST construction, new brick (colored in red) can only be placed above the pivot brick (colored in dark blue). The total number of bricks is chosen uniformly between 10 to 15. In order to obtain target images, we first transform assembled bricks to voxels in closed grid of size $32 \times 32 \times 32$ and then crop images of size $14 \times 14$ from different viewpoints with the target residing close to the center of each image.

### S.2.3   ModelNet Construction

In these experiments, we use the same subset of offset types that are available in randomly-assembled object construction whereas new brick now can be placed either above or below the pivot brick.

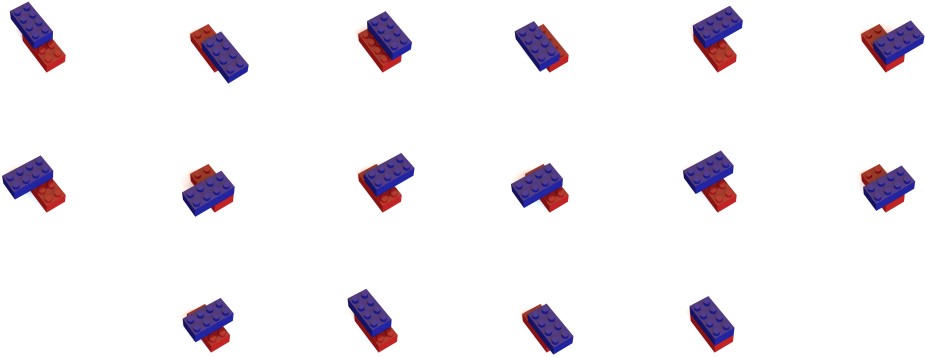

Figure s.3: Visualization of available offsets for randomly-assembled object construction.

Table s.3: Hyperparameters for MNIST construction.

| Hyperparameter | Value |
|---|---|
| Gradient clipping | 0.5 |
| Entropy coefficient | 0.01 |
| The number of timesteps | 512 |
| Total timesteps | $3 \times 10^5$ |
| The number of environments | 8 |
| Learning rate | $1 \times 10^{-4}$ |
| Gamma | 0.5 |
| Lambda | 0.9 |
| The number of epochs | 6 |
| The number of mini-batches | 32 |
| Value coefficient | 1 |

Thus, the total number of available offset types is 32, which is exactly twice of randomly-assembled object construction. The process to acquire images as the desired target information is same as in randomly-assembled object construction.

## S.3  Details of Baseline Methods and Our Method

In this section, we describe the details of baseline methods and our method $B^3$.

### S.3.1  Bayesian Optimization

We conduct Bayesian optimization [1] on the tasks we solve, following the approach proposed by Kim et al. [2]. Gaussian process regression with Matérn 5/2 kernel and expected improvement strategy are used as a surrogate function and an acquisition function. Unless otherwise specified, 5 initial points and 10 timestep budget are given for a single construction step.

### S.3.2  Supervised Learning Model

Supervised learning model is built upon the policy network of $B^3$ that is trained with the supervised learning approach instead of the reinforcement learning framework. In detail, sequence-level ground-truth of the pivot and the offset selection is used as cross entropy loss to train the network. Since value prediction of the current state is unnecessary, the value network is dropped. Due to the requirement of the pivot and the offset selection for each timestep as a label, this baseline method is only applicable in randomly-assembled object construction.

Table s.4: Hyperparameters for other benchmarks.

| Hyperparameter | Value |
|---|---|
| Gradient clipping | 0.5 |
| Entropy coefficient | 0.01 |
| The number of timesteps | 512 |
| Total timesteps | $5 \times 10^5$ |
| The number of environments | 8 |
| Learning rate | $1 \times 10^{-4}$ |
| Gamma | 0.75 |
| Lambda | 0.9 |
| The number of epochs | 6 |
| The number of mini-batches | 32 |
| Value coefficient | 1 |

Table s.5: Description of baselines and our method.

| Method | Figures | Description |
|---|---|---|
| Baseline #1 - BO | Figure 2, Figure s.4, Figure s.5 | Bayesian optimization [2] |
| Baseline #2 - SL | Figure 2(b) | Supervised learning [3] |
| Baseline #3 - MLP | Figure 2, Figure s.4, Figure s.5 | Our MLP-based method |
| Ours - GNN | Figure 2, Figure s.4, Figure s.5 | Brick-by-Brick |

### S.3.3 MLP-based Model

MLP-based model uses same pipeline as of $B^3$ but with MLPs instead of GNNs to compute features for the pivot and the offset selection. Thus, each brick feature is obtained without message passing between its neighbors. Value or estimated return of the current state, however, is computed similarly by using global average pooling over final brick features.

### S.3.4 Brick-by-Brick

Implementation details of our method $B^3$ can be found in Table s.6. The number of hidden units in both multi-layer perceptrons and convolutional neural networks is 64 if experiments are the MNIST construction task, or 192 otherwise. In both randomly-assembled object construction and ModelNet construction experiments, the dimension of target feature computed by $CNN_{tar}$ is then 192 by concatenating separately computed features of three images. The output dimension of $MLP_{piv}$ is fixed to $N_{max}$ which is 70 in ModelNet construction and 45 in the other experiments. Typically, the number of maximum bricks or the budget for target objects is predefined to values below $N_{max}$. This can be replaced to a recurrent neural network such as Pointer networks [4] if no mask information is given.

## S.4 Additional Experimental Results

All experiments are carried out on a Ubuntu 16.04 workstation, consisting of Intel(R) Core(TM) i7-6850K CPU and two NVIDIA Titan X Pascal GPUs.

Average episode return for other classes of MNIST are shown in Figure s.4. Similarly, the return curve for monitor and table classes of ModelNet are provided in Figure s.5. Note that the baseline performance is measured separately for each class of MNIST and ModelNet. We observe that $B^3$ generally outperforms the baselines in not only training episodes but also test episodes where unseen images are given. Additional qualitative results on each digit class are presented in Figure s.7. Note that three images of a constructed object in both randomly-assembled object construction and ModelNet construction are extracted from same viewpoints of the target.

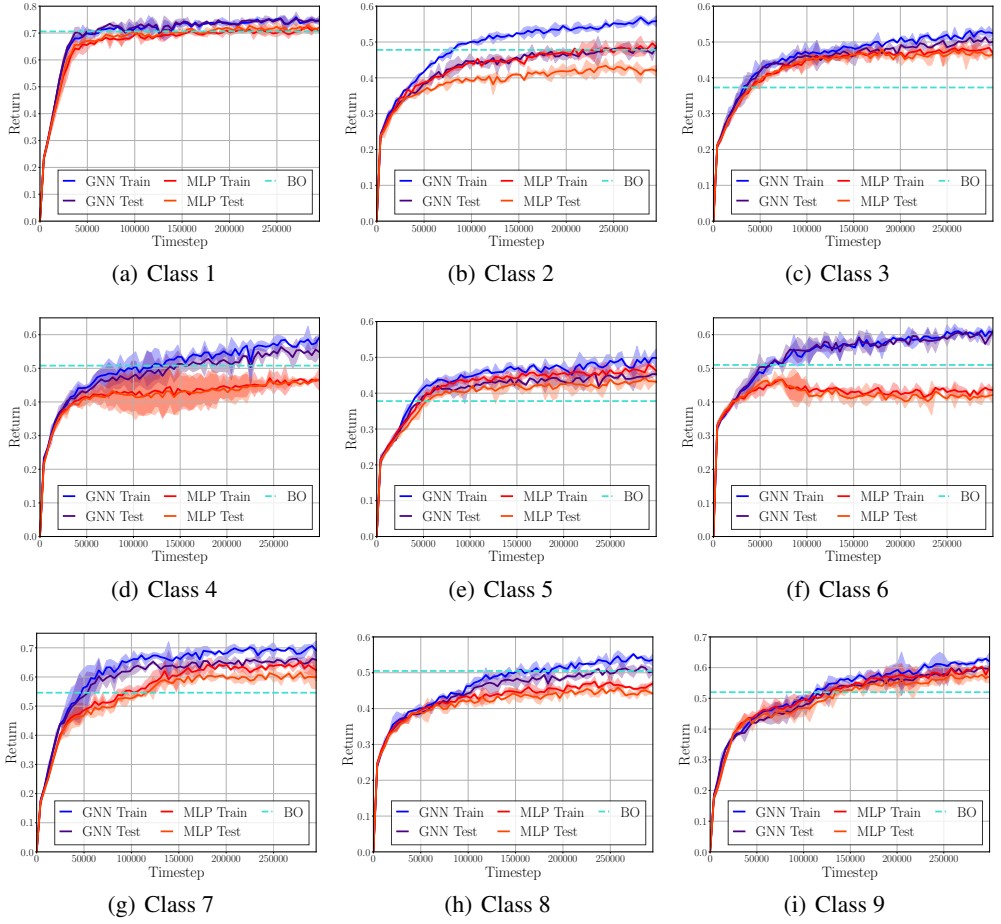

Figure s.4: Episode return curve for classes 1 to 9 in MNIST construction. Results are averaged over 3 random seeds.

## S.5 Comparison on Graph Neural Networks

We test $B^3$ on randomly-assembled object construction and compare to graph neural networks without node or edge features. Specifically, no edge model only utilizes the node features that contain positional and directional information of each brick whereas no node model only uses displacement information of edge features. The result is presented in Figure s.6. Similar to the validity prediction network experiments, $B^3$ that exploits both node and edge features reports the best performance compared to the others.

## S.6 Limitations and Societal Impacts

Our work can generate a sequence of bricks to construct a target object of which the partial information is only available. However, the partial information does not always guarantee that our model constructs a 3D object accurately because the incomplete information cannot express the object we would like to assemble. For example, the cases that belong to table category are difficult to assemble, in particular with only three views of 3D object the legs of table are not distinguishable whether a true object has two legs or four legs. This ambiguity leads us not to successfully construct a target object. To solve this problem, we can provide more information than three images from different viewpoints, but it causes an additional cost for obtaining the information. We need to balance a trade-off between elaborate information and additional cost.

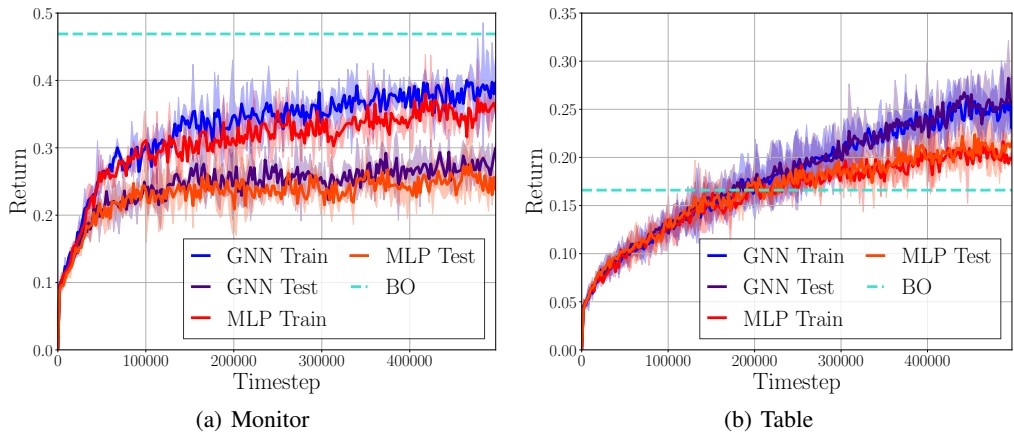

(a) Monitor            (b) Table

Figure s.5: Episode return curve for monitor and table categories in ModelNet construction. Results are averaged over 3 random seeds.

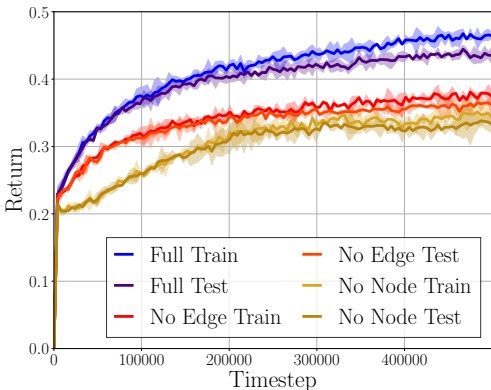

Figure s.6: Episode return curve of different GNNs for randomly-assembled object construction. Results are averaged over 3 random seeds.

If our work successfully assembles any 3D object in a combinatorial manner where partial information is given, it is capable of constructing dangerous and illegal products with basic unit primitives. For example, when 3D printing has been widely adopted, some people start to produce a dangerous and illegal object such as gun, rifle, and knife without difficulty. Similar to this, our approach can be also employed in such tasks. Additionally, due to the characteristics of combinatorial, addable, removable components, a copyright of creation is able to be easily infringed. Since our method can generate a unique sequence or assembly instruction of object diversely, the vast number of slightly different objects can be created.

Table s.6: Architecture of $B^3$. An asterisk $^*$ implies that its dimension can be changed according to a target benchmark.

| Network | Hidden Layer | Activation | Output Dimension |
|---|---|---|---|
| $\text{MLP}_v$ | FC | ReLU | $64^*$ |
| | FC | ReLU | $64^*$ |
| | FC | Linear | $64^*$ |
| $\text{MLP}_e$ | FC | ReLU | $64^*$ |
| | FC | ReLU | $64^*$ |
| | FC | Linear | $64^*$ |
| $\text{CNN}_{\text{tar}}$ | Conv2D, 32 channels, $3 \times 3$ filter, stride 1, same padding | Linear | $14 \times 14 \times 128$ |
| | Maxpool 2D, pool size 3, strides 2, same padding | ReLU | $7 \times 7 \times 128$ |
| | Conv2D, 32 channels, $3 \times 3$ filter, stride 1, same padding | ReLU | $7 \times 7 \times 128$ |
| | Conv2D, 32 channels, $3 \times 3$ filter, stride 1, same padding | ReLU | $7 \times 7 \times 128$ |
| | Conv2D, 32 channels, $3 \times 3$ filter, stride 1, same padding | ReLU | $7 \times 7 \times 128$ |
| | Conv2D, 32 channels, $3 \times 3$ filter, stride 1, same padding | Linear | $7 \times 7 \times 128$ |
| | Conv2D, 64 channels, $3 \times 3$ filter, stride 1, same padding | Linear | $7 \times 7 \times 64$ |
| | Maxpool 2D, pool size 3, strides 2, same padding | ReLU | $4 \times 4 \times 64$ |
| | Conv2D, 64 channels, $3 \times 3$ filter, stride 1, same padding | ReLU | $4 \times 4 \times 64$ |
| | Conv2D, 64 channels, $3 \times 3$ filter, stride 1, same padding | ReLU | $4 \times 4 \times 64$ |
| | Conv2D, 64 channels, $3 \times 3$ filter, stride 1, same padding | ReLU | $4 \times 4 \times 64$ |
| | Conv2D, 64 channels, $3 \times 3$ filter, stride 1, same padding | ReLU | $4 \times 4 \times 64$ |
| | Flatten | - | 1024 |
| | FC | Linear | $64^*$ |
| $\text{MLP}_v^{(\ell)}$ | FC | ReLU | $64^*$ |
| $\text{MLP}_v^{(\ell)}$ | FC | ReLU | $64^*$ |
| $\text{MLP}_{\text{piv}}$ | FC | Softmax | $N_{\text{max}}$ |
| $\text{MLP}_{\text{off}}$ | FC | Softmax | $N_{\text{off}}$ |
| $\text{MLP}_{\text{val}}$ | FC | Linear | 1 |

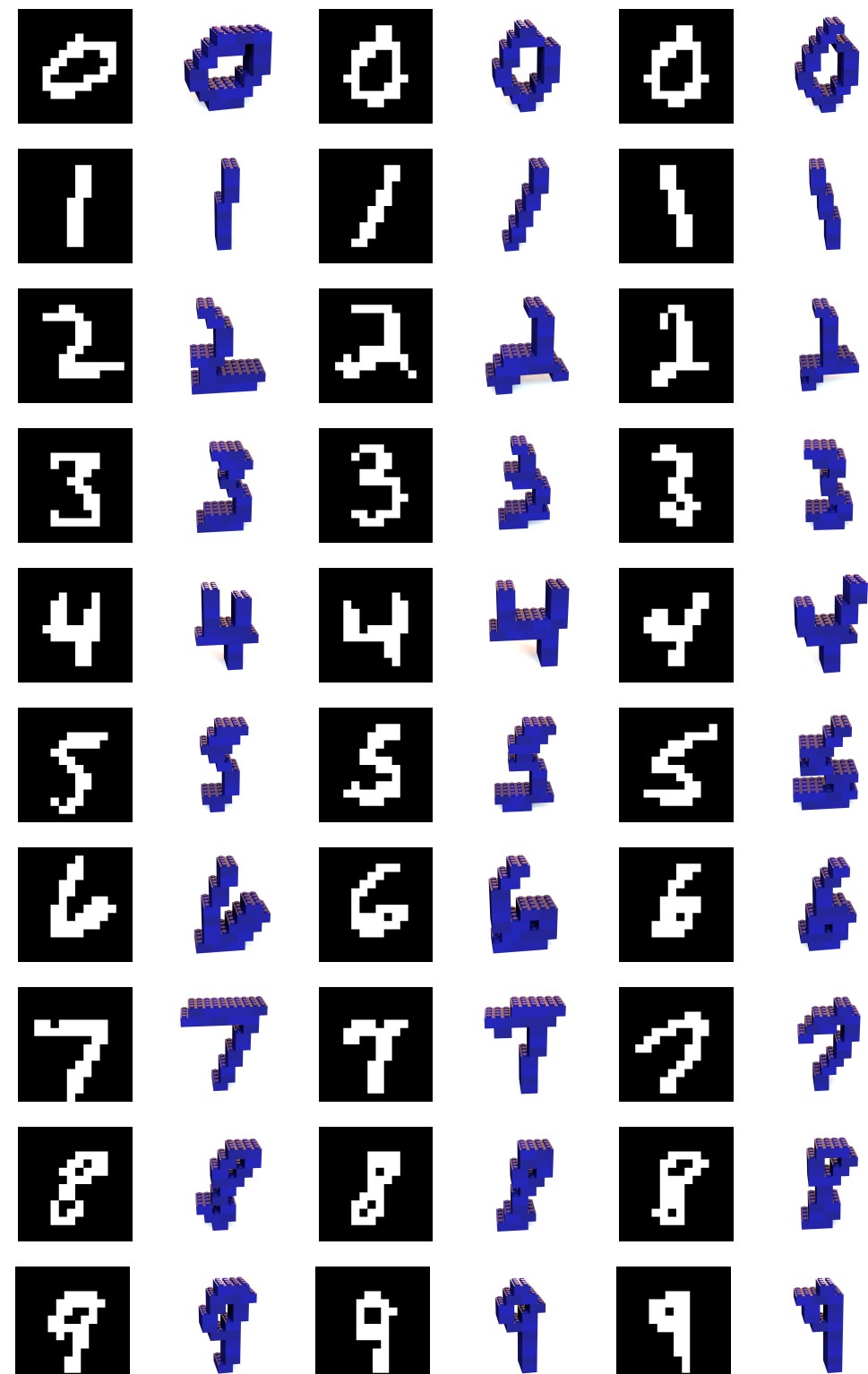

Figure s.7: Qualitative results for unseen images of all classes in MNIST construction task. The first two columns are already shown in the main article.