# OpenReview forum: "Brick-by-Brick: Combinatorial Construction with Deep Reinforcement Learning"
_NeurIPS.cc/2021/Conference — NeurIPS 2021 Poster_

### Official Review · Reviewer_cnra · 2021-07-12

**Rating:** 4
**Confidence:** 4

**Summary:**

This paper proposes a GNN+RL pipeline to use 2x4 LEGO bricks for reconstructing 3D shapes from observation of a single image or a set of images. The authors claim three main contributions: 1) the novel problem formulation of combinatorial construction; 2) the RL agent Brick-by-Brick (B^3) to solve the problem of growing action space and invalid actions in the sequential part assembly process; 3) implement an OpenAI gym environment, plus benchmark and evaluations.

**Limitations And Societal Impact:**

yes, the authors adequately addressed these issues.

**Main Review:**

**Originality:**

(positive) The paper has the technical contribution of proposing the novel problem of combinatorial construction and formulating a GNN+RL pipeline for addressing the task. Also, the authors make contributions at 1) designing the action space to make it not growing with the increasing number of bricks during assembly, by defining the action space as a tuple of <pivot brick, offset>; 2) training a neural network to predict if an action is valid or not, to reduce the growing checking time if not using the network.

(negative) However, I have some major concerns regarding the claimed contributions. Firstly, it is weird to claim "combinatorial construction" as the novel formulation proposed in this paper, as this is a general term and many previous works [19, 38] are also essentially solving the combinatorial part assembly problem. Secondly, I don't understand the necessity of training a network to predict invalid actions. If checking for O(|A_off|t^2) for an accurate answer is too computationally heavy, can you simply maintain a voxel-grid logging the brick occupancy and adding a new brick is simply a O(1)-time checking?

**Quality:**

(positive) The method design is reasonable, easy to read, and well-stated. The paper conducts a lot of experiments over different datasets.

(negative) I have many concerns regarding the experiments: 1) it seems weird to do the MNIST experiment as MNIST images are 2D shapes. Why using 3D bricks for reconstruction is an interesting task? Also, it seems that you will remove one dimension and only use a 1x2 or 1x4 2D brick to doing a 2D reconstruction, right? 2) The qualitative result for the Table shape in Fig.  6 is pretty bad. Considering that this is the only result shown for Table in the main paper, is this representing your best result? Can you achieve better results than this? Actually, this may slightly exaggerate my concern why using 2x4 bricks as the only primitive is a good choice, given such bad reconstruction quality for real ModelNet shapes? 3) Can you compare to [19, 38] as these two published papers are so related to the task in this work? In my opinion, it should be not impossible to convert them to work with image conditions. Also since your method also uses 3D ground-truth to provide rewards in Eq. 1,  I think the task settings (input/output/supervision/goal) in this paper are quite similar to [19, 38]. I thus think that a fair comparison is necessary to prove the improvement of the proposed method. 4) In table 1, it's weird to put "AVN" as a term for comparison. Do previous works not consider invalid overlapping at all? I don't think this is a big point to claim difference.

**Clarity:**

the paper is overall well-written and easy-to-read. However, the comparison to [19, 38] is not clearly described, in Sec 1/2/5.

**Significance:**

overall, I think the claimed contributions are not strong to meet NeurIPS bar, or not well-proven to be necessary. Besides, necessary detailed discussions and experimental comparisons to significantly similar works [19, 38] are missing.



**Time Spent Reviewing:**

2

---

> ### Author Response · Authors · 2021-08-10
> **Author Response to Reviewer cnra**
>
> Thank you for thoughtful comments and suggestions.
>
> * Novelty on *combinatorial construction* and the discussion on previous work including [19, 38]
>
> It is true that our work is closely related to the recent studies [19] and [38], which are preliminarily presented in NeurIPS workshops. We would like to mention that they still remain as *unpublished work* under the NeurIPS submission policy so that our work shares the novelty.
>
> As described in Part A of *Answer for All Reviewers*, Bayesian optimization-based method [19] requires *exact volumetric information* and supervised learning-based method [38] requires *ground-truth sequence* of bricks. Such settings utilize much richer information than our problem formulation, which constructs a target shape conditioned on 2D images. We will clarify our own contributions and comparisons with the work [19] and [38] in the final version.
>
> * Why do you need a separate network for action invalidation? Instead, why don’t you use the voxel logging?
>
> We have to mask out invalid actions when predicting the next action. If not, given some state, an RL agent predicts the same output in the test phase. Precisely speaking, even if the next action selected by the agent is invalid, a neural network (i.e., the RL agent) without masking out invalid actions produces the same output. For this reason, we cannot use voxel logging.
>
> Furthermore, we provided the studies without a separate network for action validation in Part B of *Answer for All Reviewers*. Please check the details in the answer.
>
> * MNIST experiments as 2D construction
>
> Yes, MNIST experiments can be considered as 2D construction. We presented these experiments as a synthetic test suite. Nevertheless, MNIST shares a common concept across digit classes (e.g., circle and line), but each class has its intrinsic characteristics (e.g., class 8 composed of two circles and class 6 composed of one circle and one line). Therefore, we think the MNIST experiments are simple but interesting.
>
> * Better results than the results in Fig. 6
>
> We have provided other results in Fig. s.7, Fig. s.8, and Fig. s.9. Please see them in the supplementary material. Additionally, we discussed the limitations of our work in Section S.6 of the supplementary material. Briefly, the information included in a set of 2D images is inherently insufficient to construct a complete 3D shape and more elaborate information induces more additional cost. We think these facts do not weaken the ability of our method, Brick-by-Brick. Therefore, we need to balance a trade-off between elaborate information and additional cost.
>
> * AVN column in Table 1
>
> Thank you for pointing out this ambiguity. Our intention was not to simply describe the existence of AVN, but we would like to categorize the type of action validation. We described the details of the potential approaches to validating actions, i.e., IMPLICIT, SAMPLING, JOINTLY, and PRETRAINED, in Part B of *Answer for All Reviewers*. We will clarify it in the final version and change AVN to *type of action validation*.
>
> [19] J. Kim, H. Chung, J. Lee, M. Cho, and J. Park. Combinatorial 3D shape generation via sequential assembly. In NeurIPS Workshop on Machine Learning for Engineering Modeling, Simulation, and Design (ML4Eng), 2020.
>
> [38] R. Thompson, G. Elahe, T. DeVries, and G. W. Taylor. Building LEGO using deep generative models of graphs. In NeurIPS Workshop on Machine Learning for Engineering Modeling, Simulation, and Design (ML4Eng), 2020.

---

> > ### Comment · Area_Chair_Vo5M · 2021-08-25
> > **Question to authors about one of the points**
> >
> > Dear Authors,
> >
> > I have a hard time understanding your answer to the reviewer's comment, could you please clarify?:
> >
> > "
> > Why do you need a separate network for action invalidation? Instead, why don’t you use the voxel logging?
> >
> > We have to mask out invalid actions when predicting the next action. If not, given some state, an RL agent predicts the same output in the test phase. Precisely speaking, even if the next action selected by the agent is invalid, a neural network (i.e., the RL agent) without masking out invalid actions produces the same output. For this reason, we cannot use voxel logging.
> > "
> >
> > If I understand correctly, the process would be: Keep track of which voxels are filled, for every available action, compute whether taking it will make new brick overlap with existing one, and if it doesn't, mask it out (e.g. subtract large number from the logits and multiply loss by 0).
> >
> > Separate question:
> > Is there any check that the resulting structure that is built won't fall apart, e.g. is not made from disconnected subsets?
> >
> > Thank you,
> > Your AC

---

> > > ### Comment · Reviewer_cnra · 2021-08-25
> > > **I'm also confused**
> > >
> > > Hi, authors,
> > >
> > > I'm also having a hard time understanding your reply. Why the RL will always predict the same answer? if it keeps putting the brick as the same location where there is already a brick, then we directly know that it's invalid, right?
> > >
> > > Best,
> > > reviewer cnra

---

> > > > ### Author Response · Authors · 2021-08-26
> > > > **Response to the question by Reviewer cnra**
> > > >
> > > > 1. Why will the RL always predict the same answer?
> > > >
> > > > By "the same answer", we meant the RL agent is deterministic at the test phase, i.e., an action corresponding to the highest probability is always chosen. To avoid an agent being stuck in choosing the same invalid action, we explore different approaches to mask invalid actions (see our results in Part B of *Answers for All Reviewers*). Simply sampling the next action until the valid one is chosen is computationally prohibitive due to a large amount of invalid actions.
> > > >
> > > > 2. If it keeps putting the brick as the same location where there is already a brick, then we directly know that it's invalid, right?
> > > >
> > > > *We provide the same response described to Area Chair Vo5M here.*
> > > >
> > > > Yes, we directly know whether the action chosen by an RL agent is invalid. However, without the separate network, the RL agent needs to keep validating an action one by one until it finds a valid action, which is time-consuming. For example, if the action chosen by the RL agent is invalid, we may be able to choose another valid action from the current state by choosing the second best action according to the action probabilities determined by the RL agent. If the action is also invalid, we need to repeat this step until we choose a valid action. This process is inefficient in the presence of a huge number of invalid actions. Furthermore, as the worst-case scenario, it needs to compute an entire ground-truth action validation in the current state. It is infeasible due to a time complexity as shown in Fig. s.1.
> > > >
> > > > As an alternative to this naive process, one may think that our RL agent is able to learn the action validity inherently without any explicit action validation network, which corresponds to IMPLICITLY in Part B of *Answer for All Reviewers*. However, it also suffers from the failure cases where it repeatedly chooses invalid actions. Please see our initial response (Part B of *Answer for All Reviewers*) for the details.
> > > >
> > > > To avoid this inefficient process and further improve other possible approaches described in Part B of *Answer for All Reviewers* (including IMPLICITLY, SAMPLING, and JOINTLY), our action validation network preemptively filters invalid actions by utilizing the PRETRAINED action validation network.
> > > >
> > > > Best regards,
> > > >
> > > > Author(s).

---

> > > ### Author Response · Authors · 2021-08-26
> > > **Response to the question**
> > >
> > > Thank you for your response.
> > >
> > > 1. Could you please clarify?: "Why do you need a separate network for action invalidation? Instead, why don't you use the voxel logging?
> > >
> > > We need the separate network mainly for efficiency. Without the separate network, the RL agent needs to keep validating an action one by one until it finds a valid action, which is time-consuming. For example, if the action chosen by the RL agent is invalid, we may be able to choose another valid action from the current state by choosing the second best action according to the action probabilities determined by the RL agent. If the action is also invalid, we need to repeat this step until we choose a valid action. This process is inefficient in the presence of a huge number of invalid actions. Furthermore, as the worst-case scenario, it needs to compute an entire ground-truth action validation in the current state. It is infeasible due to a time complexity as shown in Fig. s.1.
> > >
> > > As an alternative to this naive process, one may think that our RL agent is able to learn the action validity inherently without any explicit action validation network, which corresponds to IMPLICITLY in Part B of *Answer for All Reviewers*. However, it also suffers from the failure cases where it repeatedly chooses invalid actions. Please see our initial response (Part B of *Answer for All Reviewers*) for the details.
> > >
> > > To avoid this inefficient process and further improve other possible approaches described in Part B of *Answer for All Reviewers* (including IMPLICITLY, SAMPLING, and JOINTLY), our action validation network preemptively filters invalid actions by utilizing the PRETRAINED action validation network.
> > >
> > > 2. Is there any check that the resulting structure that is built won't fall apart, e.g., is not made from disconnected subsets?
> > >
> > > Yes, there is such a check. We consider adding a disconnected brick as an invalid action. A valid action is always connected to the structure that has already been constructed.
> > >
> > > Best regards,
> > >
> > > Author(s).

---

> > > ### Author Response · Authors · 2021-09-03
> > > **Elaborate explanation on voxel logging**
> > >
> > > Dear AC,
> > >
> > > We did not use voxel logging, because voxel logging would mask out only a tiny fraction of invalid actions. Precisely speaking, voxel logging suffers from a difficulty in handling many invalid actions including the actions that assemble a brick in the position where the voxels are empty. As mentioned in the answer for your separate question, our model disallows disconnected subsets, in order to create a constructible object. Therefore, compared to voxel logging, our approach with the action validation network can create a constructible object in the sequential manner.
> > >
> > > Best regards,
> > >
> > > Author(s).

---

> > ### Comment · Reviewer_cnra · 2021-08-25
> > **some follow-up questions**
> >
> > Hi, authors,
> >
> > thank you for your reply. Could you elaborate more?
> >
> > 1) you replied that "We have evaluated three baselines: (i) Bayesian optimization-based method [19], (ii) supervised learning-based method [38], and (iii) MLP-based agent. The results are shown in Fig. 2, Fig. s.4, Fig. s.5, and Fig. s.6". However, in these figures, there are only three algorithms, i.e. "GNN"/"MLP"/"BO"? But including your algorithm, shouldn't there be four algorithms in these figures? Also, can you link me among these names? BO is [19], GNN is [38]? then where is yours?
> >
> > 2) the shapenet reconstruction experiments take three images as inputs, right? So why only one table leg is reconstructed for example in Fig. 6(b) (similarly in the supp figures)? I don't buy your claim that "the information included in a set of 2D images is inherently insufficient to construct a complete 3D shape". Both legs are obviously visible from the front-view image, right?
> >
> > thanks

---

> > > ### Author Response · Authors · 2021-08-26
> > > **Response to the follow-up questions**
> > >
> > > Thank you for your response.
> > >
> > > 1. In these figures, there are only three algorithms. But including your algorithm, shouldn't there be four algorithms in these figures? Also, can you link me among these names?
> > >
> > > To clarify our initial response, in the table below we summarize all the four evaluated methods and link them to Figures and References. SL is only evaluated in Fig. 2b, because it requires ground-truth sequences of 3D object construction, which is only available in randomly-assembled objects.
> > >
> > > | Method            | Figures                         | Description                   |
> > > |-------------------|:-------------------------------:|:-----------------------------:|
> > > | Baseline #1 - SL  | Fig. 2b                         | Supervised learning [38]      |
> > > | Baseline #2 - BO  | Fig. 2a,b,c, Fig. s.4, Fig. s.5 | Bayesian optimization [19]    |
> > > | Baseline #3 - MLP | Fig. 2a,b,c, Fig. s.4, Fig. s.5 | Our MLP-based baseline method |
> > > | Ours - GNN        | Fig. 2a,b,c, Fig. s.4, Fig. s.5 | Our Brick-by-Brick method     |
> > >
> > > 2. The shapenet reconstruction experiments take three images as inputs, right?
> > >
> > > Yes, the ShapeNet construction experiments take three images as inputs.
> > >
> > > 3. Both legs are obviously visible from the front-view image, right?
> > >
> > > Yes, both legs are visible from the front-view image, but it is ambiguous for the model whether these legs are front or rear. In fact, a table with just three legs (i.e., one leg missing) or two legs in a diagonal direction would perfectly match with the same three input views. In addition to the fact that 2D images have insufficient information to construct a complete 3D shape, we think choosing a pivot brick for assembling legs is challenging due to a sparse reward (i.e., there are too many actions we can select) in the case of constructing a table. Note that in the case of a less ambiguous input, such as a monitor, our results are much better (Fig. s.9 in the supplementary material). We will clarify and discuss this issue by adding the results for monitor construction to Fig. 6, and comparing the table results as the failure cases that show the challenge of our problem.
> > >
> > > Please let us know if you have more questions.
> > >
> > > Best regards,
> > >
> > > Author(s).

---

### Official Review · Reviewer_2kke · 2021-07-16

**Rating:** 6
**Confidence:** 4

**Summary:**

The authors propose a combinatorial construction task consisting of assembling unit primitives (i.e. LEGOs) sequentially. The desired target is specified using one or more 2D images. The main challenges in this setting are dealing with incomplete information (of the target) and long-term planning with a variable-sized action space containing invalid actions in a combinatorially large space. To address this, the authors adopt an action validation network that filters invalid actions to an actor-critic network.

**Limitations And Societal Impact:**

Yes.

**Main Review:**

## Strengths
Constructing structures from primitives is an important task for AI and RL. The use of a 2x4 brick as the unit primitive greatly simplifies the problem, but still allows for enough complexity to be interesting and relevant.

## Weaknesses
The main technical novelty of the approach is the introduction of the action validation network, but lacks any ablation studies that demonstrate its effectiveness.

The experimental results are also a bit underwhelming. In particular, it seems like the authors are training on single classes individually for each of the datasets (e.g. a single digit for MNIST). It is not clear why this is done, and would be very limiting if the training could not be done more generally. Additionally, besides the ablations, only one baseline is tested, a Bayesian optimization-based approach (BO). However, since BO requires exact volumetric information it would be helpful to include others that are more comparable.

## Correctness
The methodology seems correct to the best of my knowledge.

## Clarity
Writing is clear.

## Relation to Prior Work
Prior work on assembly is discussed, but it is unclear why none were used as baselines in this work (see weaknesses).

## Additional Feedback


**Time Spent Reviewing:**

3

---

> ### Author Response · Authors · 2021-08-10
> **Author Response to Reviewer 2kke**
>
> Thank you for thoughtful comments and suggestions.
>
> * Lacks of ablation studies on action validation network
>
> We provided the ablation studies on the action validation network. Please see Part B of *Answer for All Reviewers*.
>
> * Baselines including Bayesian optimization-based method
>
> As described in Part A of *Answer for All Reviewers*, we have already tested three baselines: (i) Bayesian optimization-based method [19], (ii) supervised learning-based method [38], and (iii) MLP-based agent. Please check our answer.
>
> [19] J. Kim, H. Chung, J. Lee, M. Cho, and J. Park. Combinatorial 3D shape generation via sequential assembly. In NeurIPS Workshop on Machine Learning for Engineering Modeling, Simulation, and Design (ML4Eng), 2020.
>
> [38] R. Thompson, G. Elahe, T. DeVries, and G. W. Taylor. Building LEGO using deep generative models of graphs. In NeurIPS Workshop on Machine Learning for Engineering Modeling, Simulation, and Design (ML4Eng), 2020.

---

### Official Review · Reviewer_ataC · 2021-07-16

**Rating:** 7
**Confidence:** 4

**Summary:**

This paper proposes a new task of constructing a 3D object by assembly of Lego-like primitive bricks, given only image/s as the target. They propose an RL agent that by incorporating Graph neural networks and action validation module is able to tackle the proposed task. They demonstrate the agent’s ability on three different sets of data.

**Limitations And Societal Impact:**

Yes.

**Main Review:**

The proposed problem is very interesting and hard due to its combinatorial properties, but I think it would be useful if the authors could elaborate more on the justification of why this is an interesting problem, how it could have a broader impact, and potential applications.

Some of the methods introduced in the paper such as the issue of dealing with variable and large action space are a great contribution beyond the use case in the current work.

I do not think there is much evidence that humans learn about objects by decomposing objects into very primitive elements. Rather the segmentation is done semantically. Would be interested to be pointed to literature that demonstrates otherwise.

Section 2. An illustration of brick in this section could help the reader better understand the text. For example in L66: an illustration of some of the combinations of two bricks could be very informative for the reader to visualize the combinatorial space. Same for the offset locations on a brick.

L115: Is x_t in a quantized space due to the nature of the bricks?

What happens if the action validation network is wrong and the proposed action is invalid?

Most of the experimental comparisons are more like ablation of the B3. Would it be possible to adopt some of the prior work such as those listed in Table 1 for this task and show comparison results with them?

**Time Spent Reviewing:**

4

---

> ### Author Response · Authors · 2021-08-10
> **Author Response to Reviewer ataC**
>
> Thank you for your thoughtful comments and suggestions.
>
> * Elaborate explanation for justification of why this is an interesting problem, a broader impact, and potential applications
>
> The problem we solved in this paper closely depicts how humans understand an object. Humans naturally analyze a 3D object by picturing its part-by-part decomposition and consequently grasp a rich semantic understanding.
>
> In the perspective of technical difficulty, our problem is very challenging due to the nature of *combinatorial structures* and the existence of *invalid actions*. Similar to traveling salesperson and minimum spanning tree, our problem is defined on a combinatorial space, characterized by discrete variables and their combinations.
>
> We would like to emphasize that our framework on combinatorial construction is capable of creating an instruction for assembling a 3D shape using structured primitives, e.g., bricks in our case, conditioned on 2D images. This is applicable to a real-world construction problem where structured primitives are often used; existing 3D shape generation (construction) approaches using meshes and point clouds are very limited in this practical sense. We showed that reinforcement learning is an appropriate approach compared to the work [19] and [38] in this problem. Furthermore, in the perspective of technical novelty, the existence of a huge number of invalid actions makes our problem more challenging. Our action validation network helps us to train an RL agent with coping with invalid actions.
>
> Potential applications are additive manufacturing with primitives, infrastructure construction, and microstructure design. We think our problem formulation helps us to efficiently solve such problems. Moreover, a different view of our problem is described in Section S.6.
>
> * Evidence that humans learn about objects by decomposing objects into very primitive elements
>
> As discussed in the work [R-3] and [R-4], visual object understanding is derived from diverse cognitive concepts including recognition-by-components. Recognition-by-components implies that an object is represented with a small number of three-dimensional primitives called GEONS. By this understanding, evidence that humans learn about objects by decomposing objects into very primitive elements can be supported.
>
> * Illustration of brick and possible combinations
>
> Thank you for suggesting this idea. We have visualized them in Fig. s.3 and Fig. s.4, but we will move them to our main paper and add more explanation.
>
> * Is $x_t$ in a quantized space due to the nature of the bricks?
>
> Yes, bricks are placed on a quantized space.
>
> * What happens if the action validation network is wrong and the proposed action is invalid?
>
> The episode that selects an invalid action is then terminated. If we assume that our approach can determine a valid action without an additional component for action validation (IMPLICIT baseline as described in Part B of *Answer for All Reviewers*), there exist a huge number of the episodes that are early terminated due to invalid action selection, which greatly reduces training efficiency; in our experience we cannot successfully train the agent using this scheme.
>
> * Baselines listed in Table 1
>
> We described them in Part A of *Answer for All Reviewers*. Additionally, as the method of the work [3] solves a different task with a sticky point, it cannot be used as a baseline for our work. However, we have considered their architecture design into our proposed model Brick-by-Brick.
>
> [3] V. Bapst, A. Sanchez-Gonzalez, C. Doersch, K. Stachenfeld, P. Kohli, P. Battaglia, and J. Hamrick. Structured agents for physical construction. In Proceedings of the International Conference on Machine Learning (ICML), pages 464-474, 2019.
>
> [19] J. Kim, H. Chung, J. Lee, M. Cho, and J. Park. Combinatorial 3D shape generation via sequential assembly. In NeurIPS Workshop on Machine Learning for Engineering Modeling, Simulation, and Design (ML4Eng), 2020.
>
> [38] R. Thompson, G. Elahe, T. DeVries, and G. W. Taylor. Building LEGO using deep generative models of graphs. In NeurIPS Workshop on Machine Learning for Engineering Modeling, Simulation, and Design (ML4Eng), 2020.
>
> [R-3] I. Biederman. Recognition-by-Components: A Theory of Human Image Understanding. Psychological Review, 94(2), page 115, 1987.
>
> [R-4] T. J. Palmeri and I. Gauthier. Visual Object Understanding. Nature Reviews Neuroscience, 5(4), pages 291-303, 2004.

---

### Official Review · Reviewer_n1WS · 2021-07-26

**Rating:** 5
**Confidence:** 3

**Summary:**

The proposed combinatorial construction is a new formulation for combinatorial problems
which have a vast number of possible combinations. This requires the agent to assemble LEGO
units sequentially to build the desired structure based on the 2D target image. The authors use a
reinforcement learning method with Graph-Neural-Network to solve this formulated problem
which requires a comprehensive understanding of the incomplete input information and long-
term planning. The authors also introduce an action validation network to help eliminate
invalid actions when the action space increases as the assembly processes. The
proposed method is evaluated on three types of tasks with different inputs and is compared
with three types of baselines: Bayesian optimization-based method, fully-MLP architecture, and supervised learning method.

**Ethical Concerns:**

No ethical concerns.

**Limitations And Societal Impact:**

The proposed propose an interesting formulation of one combinatorial problem and
implement this in the OpenAI Gym environment. Although, the proposed method works well
on this combinatorial construction problem and outperforms some baselines, the paper lacks
some necessary experiments to fully support the usage the action validation network.

Also, the motivation of the problem setup does not seem to be well justified, such as the use
of images instead of 3D models, and why only DRL is picked as baselines? What about other
type of baseline solutions such as search-based method, or even supervised learning?

**Main Review:**

*Strengths*:
First of all, the paper is well organized and written, and the reader can easily follow the logic of the
paper. The proposed method has a good generalization ability among different types of 2D
inputs: single view and multiple views. The experiment results show the proposed RL based
method outperforms the fully-MLP based method which indicated the effectiveness of using
GNN architecture for embedding the input data. From the qualitative results on both MINST
and ModelNet, it is shows that the proposed promise a plausible 3D construction with the
partial information.

*Weakness*:
1. The proposed formulation for this combinatorial problem is interesting, but I don’t see it
would be easily transfer to real construction problems, especially when the authors claims that they
"focus on the real-world construction procedure". Because it is hard to use the
action validation network to eliminate the invalid actions when the action space will be
continuous in real world problem.
2. The action validation network is pre-trained when used in the experiments. It is
necessary to show how the performance would be when the action validation network is
trained jointly with the reinforce pipeline.
3. It is also interesting to show some ablation study on this action validation network.
4. Would the performance be improved when using a more complete 3D model as input for
the proposed method? Is it reasonable to assume only images as input, given that 3D models are available?
5. There are also some concerns about the baseline selection and justification (more on this below).

**Time Spent Reviewing:**

2 hours

---

> ### Author Response · Authors · 2021-08-10
> **Author Response to Reviewer n1WS**
>
> Thank you for your thoughtful comments and suggestions.
>
> * Action validation network on a continuous space
>
> An action validation network defined on a continuous space is out of the scope of our work because our current method solves the problem on a discrete space. However, it is an interesting direction indeed and we may be able to extend our research in that direction in future. A possible approach would be to implicitly learn the next valid action, similar to what we described in *Answer for All Reviewers*.
>
> * Jointly trained action validation network
>
> Thank you for the suggestion. We have tested the scenario. Please see Part B of *Answer for All Reviewers*.
>
> * Ablation studies on action validation network
>
> We have provided the studies on the action validation network. Please see Part B of *Answer for All Reviewers*.
>
> * Is it reasonable to assume only images as input, instead of a more complete 3D model?
>
> In terms of real-world scenarios, 2D images are significantly easier to get and also more popular for visual data than 3D models such as meshes or point clouds. Furthermore, humans also tackle the problem of assembly by observing 2D images of the target only without  access to the full 3D model information. Some previous work on 3D reconstruction, the work [R-1] and [R-2], also considers a setting similar to our problem formulation.
>
> * Lacks of baselines including search-based method and supervised learning-based method
>
> We have already tested a search-based method (e.g., Bayesian optimization [19]) and supervised learning-based (e.g., supervised learning with ground-truth sequence [38]). Please see Part A of *Answer for All Reviewers*.
>
> [19] J. Kim, H. Chung, J. Lee, M. Cho, and J. Park. Combinatorial 3D shape generation via sequential assembly. In NeurIPS Workshop on Machine Learning for Engineering Modeling, Simulation, and Design (ML4Eng), 2020.
>
> [38] R. Thompson, G. Elahe, T. DeVries, and G. W. Taylor. Building LEGO using deep generative models of graphs. In NeurIPS Workshop on Machine Learning for Engineering Modeling, Simulation, and Design (ML4Eng), 2020.
>
> [R-1] H. Xie, H. Yao, X. Sun, S. Zhou, S. Zhang. Pix2Vox: Context-Aware 3D Reconstruction From Single and Multi-View Images. In Proceedings of the IEEE/CVF International Conference on Computer Vision (ICCV), pages 2690-2698, 2019.
>
> [R-2] T. Hu, L. Wang, X. Xu, S. Liu, J. Jia. Self-Supervised 3D Mesh Reconstruction From Single Images. In Proceedings of the IEEE/CVF Conference on Computer Vision and Pattern Recognition (CVPR), pages 6002-6011, 2021.

---

### Author Response · Authors · 2021-08-10
**Answer for All Reviewers**

Dear all reviewers,

We appreciate your valuable comments.

First of all, we would like to clarify two common issues: (i) comparisons with previous work and (ii) analysis on the action validation network.

# A. Comparisons with previous work

Some reviewers appear to think that we did not evaluate important baselines, in particular the work [19] and [38], but we believe there is a misunderstanding here. We have evaluated three baselines: (i) Bayesian optimization-based method [19], (ii) supervised learning-based method [38], and (iii) MLP-based agent. The results are shown in Fig. 2, Fig. s.4, Fig. s.5, and Fig. s.6 (*s* indicates that it is located in the supplementary material).

For the experiments, we used the implementations of the work [19] and [38], which were provided by the authors of the work [19] and [38]. However, as shown in Table 1 and also pointed out by **Reviewer 2kke**, the method of the work [19] requires *exact volumetric information*, which uses much richer information than Brick-by-Brick. Moreover, the work [38] requires a *ground-truth sequence* to construct a 3D object, which makes the method very limited; see the discussion in the work [38] for more details. For this reason, it is impossible to test the supervised learning-based method [38] on MNIST and ModelNet test suites.

Furthermore, to verify the importance of graph-structured input (see Section S.3.2 and Section S.3.4 in the supplementary material), we have also tested an MLP-based agent as shown in Fig. 2, Fig. s.4, and Fig. s.5. The results show that our method, Brick-by-Brick, outperforms this baseline.

Additionally, we have validated the effects of graph components (i.e., nodes and edges) in Fig. s.6. The model using all the components shows better performance than the model  without node or edge components; please see Section S.5 in the supplementary material for more details.

# B. Analysis on action validation network

Some reviewers requested ablation studies and further analysis on the action validation network. To study the effects of the action validation network in depth, we introduce three baseline methods and compare them with our action validation network.

1. *IMPLICIT* baseline: determining a valid action without an additional component for action validation (e.g., the work [38]):

An agent can learn how to distinguish which action is valid, by implicitly entrusting the RL agent with this ability. Although the work [38] did not employ an RL framework, a method to cope with valid or invalid action, partially proposed by the work [38], can be categorized as IMPLICIT.

2. *SAMPLING* baseline: sampling a fixed-sized set of valid actions and choosing one from the sampled action set:

An agent samples a fixed-sized set of valid actions and chooses one action from the sampled set. Bayesian optimization-based method [19] adopts this technique to distinguish between valid and invalid actions.

3. *JOINTLY* baseline: determining a valid action with a component for action validation, but the component is jointly trained with an RL agent.

It has an explicit component for action validation, but it is simultaneously trained with an RL agent. Since action validation requires the complexity $\mathcal{O}(|A_{off}|t^2)$, this approach increases training time consumed in the training phase. Moreover, the component cannot be reused across different tasks.

4. *PRETRAINED* (ours): determining a valid action with a component for action validation that can be pretrained (e.g., *ours*)

Such an approach has a component for action validation explicitly and pretrains it. This component helps us to determine a valid action with a single forward pass and reuse the component across different tasks. This corresponds to our action validation network.

We summarize the properties of the methods above in the following table.

| Method | Separate component | No access to action validity in test phase | Reusability |
|--------|:------------------:|:------------------------------------------:|:-----------:|
| IMPLICITLY        |                    | V |                    |
| SAMPLING          | V |                    |                    |
| JOINTLY           | V | V |                    |
| PRETRAINED (Ours) | V | V | V |

From now, we provide quantitative studies on the four methods. To fairly compare all the methods, we use the same training and test datasets of randomly-assembled objects, which are described in Section 4 of our paper, and an RL agent produces the next action from a dataset. SAMPLING samples 1,500 actions from a set of all the actions including valid and invalid actions.

1. Training phase

| Method | Precision for Pivot | Recall for Pivot | Precision for Offset | Recall for Offset | Reward |
|--------|:-------------------:|:----------------:|:--------------------:|:-----------------:|:------:|
| IMPLICITLY        | N/A        | N/A        | N/A        | N/A        | 0.3752     |
| SAMPLING          | N/A        | N/A        | N/A        | N/A        | 0.9769     |
| JOINTLY           | 0.9881     | **0.9988** | 0.9001     | 0.9505     | 0.9494     |
| PRETRAINED (Ours) | **0.9969** | **0.9988** | **0.9212** | **0.9892** | **0.9880** |

2. Test phase

| Method | Precision for Pivot | Recall for Pivot | Precision for Offset | Recall for Offset | Reward |
|--------|:-------------------:|:----------------:|:--------------------:|:-----------------:|:------:|
| IMPLICITLY        | N/A        | N/A        | N/A        | N/A        | 0.3491     |
| SAMPLING          | N/A        | N/A        | N/A        | N/A        | 0.8672     |
| JOINTLY           | 0.9809     | **0.9982** | 0.8674     | 0.9467     | 0.9450     |
| PRETRAINED (Ours) | **0.9905** | 0.9955     | **0.8904** | **0.9868** | **0.9824** |

PRETRAINED shows the best performance than IMPLICITLY, SAMPLING, and JOINTLY. In particular, both IMPLICITLY and SAMPLING struggle to assemble an object. JOINTLY is better than both IMPLICITLY and SAMPLING, but it is lower than PRETRAINED. Diverse episodes, generated from an environment, does not help to learn a generalization of how to distinguish between valid and invalid actions.

Here, we discuss the time complexity of action validation methods. Because wall-clock time is unreliable to measure the time complexity due to the bottleneck from other components of RL agent, we present the ranks of methods.

* IMPLICITLY $\lessapprox$ JOINTLY $=$ PRETRAINED $\ll$ SAMPLING

IMPLICITLY does not need to compute action validation. On the other hand, both JOINTLY and PRETRAINED are slightly slower than IMPLICITLY, due to the existence of action validation networks. SAMPLING is the slowest method, because it always requires access to action validity.

Additionally, we have visualized *the time complexity of computing ground-truth action validation* in Fig. s.1 of the supplementary material. Such a figure shows that considering ground-truth action validation in an online manner is infeasible in practice.

[19] J. Kim, H. Chung, J. Lee, M. Cho, and J. Park. Combinatorial 3D shape generation via sequential assembly. In NeurIPS Workshop on Machine Learning for Engineering Modeling, Simulation, and Design (ML4Eng), 2020.

[38] R. Thompson, G. Elahe, T. DeVries, and G. W. Taylor. Building LEGO using deep generative models of graphs. In NeurIPS Workshop on Machine Learning for Engineering Modeling, Simulation, and Design (ML4Eng), 2020.

Best regards,

Author(s)

---

> ### Author Response · Authors · 2021-08-27
> **Clarification of our initial response on baseline methods**
>
> As requested by **Reviewer cnra**, we summarize all the four evaluated methods and link them to Figures and References as shown in the following table. To clarify our initial response, we also post the table here. SL is only evaluated in Fig. 2b, because it requires ground-truth sequences of 3D object construction, which is only available in randomly-assembled objects.
>
> | Method            | Figures                         | Description                   |
> |-------------------|:-------------------------------:|:-----------------------------:|
> | Baseline #1 - SL  | Fig. 2b                         | Supervised learning [38]      |
> | Baseline #2 - BO  | Fig. 2a,b,c, Fig. s.4, Fig. s.5 | Bayesian optimization [19]    |
> | Baseline #3 - MLP | Fig. 2a,b,c, Fig. s.4, Fig. s.5 | Our MLP-based baseline method |
> | Ours - GNN        | Fig. 2a,b,c, Fig. s.4, Fig. s.5 | Our Brick-by-Brick method     |
>
> Best regards,
>
> Author(s)

---

### Author Response · Authors · 2021-09-01
**Thank you for your comments again**

Dear Reviewers,

We sincerely appreciate your efforts in providing helpful comments.

We did our best to address your concerns in our rebuttal.

Please check it out and update your review.

We are also happy to respond if you have any follow-up questions.

Best regards,

Author(s).

---

### Decision · Program_Chairs · 2021-09-27

**Decision:**

Accept (Poster)

**Comment:**

The paper addresses a problem a construction a 3d object with lego, in the action space of locations of placement of a new block, purely from visual observation during test time and trained with reinforcement learning to match the volume of the target object.

The main positives of this paper are that this is an interesting setup/problem that has not been addressed before as far as we can tell (the lego construction yes, but from purely visual, trained with RL), they propose an interesting solution of graph neural network with action validation network and succeed at building basic shapes.

The main drawback is that while they do succeed in making 3d shape objects (like an airplane), the quality of the constructed objects is not very high and the objects are not so complex (e.g. don’t have that many blocks). However, as this has not been done before, it’s hard to know what to expect - how difficult this problem really is.

Because of this and because the setup is novel and this topic is quite uncommon and yet interesting, important and should be studied more, I think the paper should be accepted.

For the authors I primarily recommend pushing the performance - getting larger, more diverse 3d shapes constructed of at higher level of quality.

There were also few things unclear in the discussion, I also recommend for the authors to explain them with better writing.